# In Vivo Cellular Phosphatidylcholine Kinetics of CD15+ Leucocytes and CD3+ T-Lymphocytes in Adults with Acute Respiratory Distress Syndrome

**DOI:** 10.3390/cells13040332

**Published:** 2024-02-11

**Authors:** Ahilanandan Dushianthan, Rebecca Cusack, Victoria Goss, Grielof Koster, Michael P. W. Grocott, Anthony D. Postle

**Affiliations:** 1Perioperative and Critical Care Theme, NIHR Southampton Biomedical Research Centre, University Hospital Southampton NHS Foundation Trust, Southampton SO16 6YD, Hampshire, UK; r.cusack@soton.ac.uk (R.C.); g.koster@soton.ac.uk (G.K.); mike.grocott@soton.ac.uk (M.P.W.G.); a.d.postle@soton.ac.uk (A.D.P.); 2Integrative Physiology and Critical Illness Group, Clinical and Experimental Sciences, Faculty of Medicine, University of Southampton, Southampton SO16 6YD, Hampshire, UK; 3Clinical Trials Unit, University Hospital Southampton NHS Foundation Trust, Southampton SO16 6YD, Hampshire, UK; v.m.goss@soton.ac.uk

**Keywords:** ARDS, neutrophil, lymphocyte, phospholipids, phosphatidylcholine, *methyl-*D_9_-choline

## Abstract

Mammalian cell membranes composed of a mixture of glycerophospholipids, the relative composition of individual phospholipids and the dynamic flux vary between cells. In addition to their structural role, membrane phospholipids are involved in cellular signalling and immunomodulatory functions. In this study, we investigate the molecular membrane composition and dynamic flux of phosphatidylcholines in CD15+ leucocytes and CD3+ lymphocytes extracted from patients with acute respiratory distress syndrome (ARDS). We identified compositional variations between these cell types, where CD15+ cells had relatively higher quantities of alkyl-acyl PC species and CD3+ cells contained more arachidonoyl-PC species. There was a significant loss of arachidonoyl-PC in CD3+ cells in ARDS patients. Moreover, there were significant changes in PC composition and the *methyl*-D_9_ enrichment of individual molecular species in CD15+ cells from ARDS patients. This is the first study to perform an in vivo assessment of membrane composition and dynamic changes in immunological cells from ARDS patients.

## 1. Introduction

The specificity of the membrane phospholipid composition for different cell types is highly regulated in vivo, depends on a balance between phenotypic expression and cellular nutrition and is tightly linked to cell function [1]. Examples include the synthesis and secretion of disaturated phosphatidylcholine (PC) by the lung alveolar epithelium, the critical role of tetralioleoyl cardiolipin in muscle mitochondrial function and the importance of stearoylarachidonoyl phosphatidylinositol (PI) and related polyphosphates in cellular signalling [2,3,4]. Despite a common nutritional exposure, circulating leukocytes exhibit a considerable divergent membrane phospholipid composition. Peripheral blood mononuclear cells (PBMCs) from healthy adults have relatively high abundances of arachidonoyl-containing PC and PI species combined with an elevated fractional abundance of dipalmitoyl PC [5]. By contrast, polymorphonuclear cells (PMNCs) are characterised by lower amounts of arachidonoyl PC and PI with higher fractional abundances of monounsaturated and alkylacyl molecular species [5]. 

We have previously reported that PC species synthesised by isolated PMNCs ex vivo through the CDP-choline pathway, undergo significant acyl remodelling by the Lands pathway, with the synthesis of arachidonoyl-PC species from deuterated *methyl*-D_9_-choline being double the equilibrium composition [5]. The same study reported minimal acyl remodelling by PBMCs. Given the importance of eicosanoid mediators generated by PMNCs from membrane-derived arachidonate to potentiate local inflammation [6], an increased understanding of the mechanisms regulating the arachidonic acid incorporation into the membrane phospholipids in these cells is important. Moreover, as the viability of isolated PBMCs in vitro is very limited, such mechanisms also need to be demonstrated in vivo.

Acute respiratory distress syndrome (ARDS) is associated with severe hypoxemia and carries a significant mortality with an associated increased economic burden [7]. The pathophysiology of ARDS is characterised by neutrophil-mediated inflammatory responses, leading to alveolar epithelial and endothelial injury [8]. More recently, the neutrophil-to-lymphocyte ratio (NLR) has emerged as a key marker of systemic inflammation and has potential as a prognostic indicator for clinical outcomes in patients with severe COVID-19 [9]. The regulation of inflammation is crucially controlled by a variety of cell membrane phospholipids and their downstream metabolites, including eicosanoids [10]. We recently demonstrated a gradual loss of arachidonate-PC in plasma and erythrocytes from patients with ARDS, and a comparable loss in PBMCs may cause a variable degree of substrate availability for the synthesis of inflammatory mediators [11]. The characterisation of the cell membrane phospholipid composition and de novo synthesis may improve the understanding of the underlying mechanisms of the complex phospholipid flux between cells. However, such studies of in vivo human models exploring pathological conditions are lacking. 

In this study, we assessed the PC composition in peripheral CD3+ and CD15+ leukocytes with the additional characterisation of the dynamic turnover of individual molecular PC species in CD15+ cells, in healthy volunteers and during the early stages of ARDS. Stable isotope labelling with *methyl*-D_9_-choline in combination with the analytical methods of electrospray ionisation mass spectrometry, enabled the assessment of dynamic PC flux through human CD15+ leucocytes with direct comparisons to the CD3+ cellular PC molecular composition in vivo.

## 2. Materials and Methods

### 2.1. Human Participant Selection

The study was approved by the National Research Ethics Committee (10/WNo01/52 and 11/SC/0185) and University Hospital Southampton Research and Development Department. After consent or assent patients with ARDS and healthy volunteer controls underwent an intravenous infusion of *methyl*-D_9_ choline chloride at 3.6 mg/kg for 3 h. Two ml of blood samples were collected in EDTA specimen bottles at base line and 6, 12, 24, 48, 72 and 96 h from patients and at baseline and 8, 24, 48, 72 and 96 h after the *methyl*-D_9_ choline infusion. The controls were healthy volunteers without any prior existing medical problems.

### 2.2. CD15+ Leucocytes and CD3+ Cell Isolation

CD15+ leucocytes were extracted from whole blood with CD15+ and CD3+ microbeads using the autoMACS system (Miltenyi Biotec, Tokyo, Japan). In total, 1 mL of venous blood sample was mixed with 50 μL of either CD15 or CD3 microbeads and was incubated for 15 min at 4 °C. The cells were then washed with 4 mL of autoMACS running buffer (phosphate-buffered saline, pH 7.2 + 2 mM EDTA+ 0.5% BSA+ 0.05% of *v*/*v* sodium azide) and centrifuged at 400× *g* for 10 min at room temperature. The supernatant was discarded, and the resulting cell pellet was re-suspended in 2 mL of autoMACS running buffer. The samples were then passed through a high-density magnetic cell separator, and specific cells were eluted as the positive fraction. The eluted positive faction was centrifuged at 400× *g* and 4 °C for 10 min. A visible cell pellet was obtained by discarding the supernatant.

### 2.3. Phospholipid Extraction

A modified Bligh and Dyer method was applied for the extraction of phospholipids [12]. Each sample was made up to 800 μL with the addition of normal saline. In total, 1 nM of dimyristoylphosphatidylcholine (PC 14:0/14:0) was added as the internal standard. Chloroform/methanol/water (2:2:1 *v*/*v*) was added and then vortexed vigorously and centrifuged at 445× *g* for 20 min. The lower phospholipid-rich layer was carefully aspirated and dried completely at 37 °C through a nitrogen gas concentrator. Once dried, the sample was re-dissolved further in 1 mL of chloroform and dried again to be analysed via ESI-MS.

### 2.4. Nanoflow Electrospray Ionisation Mass Spectrometry (ESI-MS/MS)

The dried phospholipid fraction was suspended in 20 µL of a mixture of methanol-butanol-water-NH_4_OH (6:2:1.6:0.4 *v*/*v*) to be analysed via electrospray ionisation mass spectrometry (Micromass, Wythenshawe, UK). Samples were injected via nanoflow spray to enhance intensity and specificity. The capillary voltage was set at 1.25 kV with a cone voltage of 30 V. Phospholipids were measured via fragment scanning. Unlabelled and *methyl*-D_9_-labelled PC molecules were measured alternatingly in the same scan using precursor scans of P184+ and P193+ using the same conditions, respectively. From sequential precursor scans of the m/z P184+ and P193+, it is possible to assess the compositions of endogenous and labelled PC species. PC masses were scanned between 670 and 870. 

Nanoflow ESI-MS/MS readily resolved PC and sphingomyelin species as precursor scans of *m/z* 184+ and typical mass spectra for positively eluted CD3+ and CD15+ cells from a healthy control are presented in Figure 1. Fourteen major PC species were identified with a fractional composition of >1% total abundance that were common to both cell types and were included in the data analysis. Due to the low number of cells, especially in the control group, longer acquisition times were required to generate spectra for the incorporation of *methyl-*D_9_ choline, and this precluded the analysis of additional phospholipid classes. Additionally, the smaller mass of CD3+ cells means that the ESI-MS signal from the *methyl*-D_9_choline incorporation was too unreliable for the estimation of their PC synthetic rates.

### 2.5. Data Extraction and Statistical Analysis

Dedicated excel spread sheets were used to quantify ion peaks for the endogenous and labelled species to calculate the turnover of phosphatidylcholines. Data were summarised as the means and standard errors of the mean. The difference in composition in each group was examined by calculating the differences in the means using the Student’s *t*-test for unpaired or paired data as appropriate. Statistical significance was assumed when the *p* value was <0.05.

## 3. Results

### 3.1. Participant Characteristics

Nine mechanically ventilated ARDS patients enrolled into the study. The median age was 62 (range 38–90) with a mean PaO_2_/FiO_2_ ratio of 108.8 ± 1.4 mmHg. Murray’s mean lung injury score and lung compliance on admission was 3.1 ± 0.1 and 36.0 ± 6.1 cmH_2_O, respectively. All patients were enrolled within 72 h of ARDS diagnosis. The majority had pneumonia as a precipitating cause for their ARDS. Five controls were healthy volunteers, not age-matched, and had a median age of 26 (range 18–36) with no prior medical problems or history of smoking. The *methyl-*D_9_ choline chloride infusion dose was similar between ARDS patients (274 ± 34 mg) and healthy controls (270 ± 24 mg).

### 3.2. CD3+ and CD15+ Cellular PC Composition in Healthy Humans

While PC in CD3+ and CD15+ cells from healthy volunteers were composed of similar PC species, their individual relative proportions were significantly different. CD15+ cell PC consisted mainly of diacyl mono- (33.2%) and di-unsaturated (22.4%) species, followed by alkyl-acyl (28%) and, to a lesser extent, polyunsaturated species (10.4% total PC). The contribution of putative arachidonoyl-containing species was very low (PC36:4 1.8%, PC38:4 2.2%, PC38:5 1.3%). By contrast, CD3+ cell PC was considerably more enriched in arachidonoyl species, but with a much smaller proportion of ether-linked PC species (Figure 2).

### 3.3. CD15+ Cellular PC Composition in ARDS

ARDS patients exhibited significant changes in cell PC composition, which were largely cell-type specific. These changes were restricted to a limited number of molecular species for CD15+ cells, with decreased PC36:1 and selected alkylacyl species accompanied by an increased PC36:2 but no alteration to the abundance of arachidonoyl PC species with no significant time course variations during the 96 h study period (Figure 3).

Overall, in the CD15+ cells from ARDS patients, there were increments in the pool of total di-unsaturated PC species and polyunsaturated PC species with significant but small decreases in monounsaturated and 1-alkyl PC species (Figure 4).

### 3.4. CD3+ Cellular PC Composition in ARDS

By contrast, compositional changes were more widespread for CD3+ cells, characterised by decreased contributions in the arachidonoyl and disaturated species (PC34:4, PC36:4, PC36:5, PC32:0) and increased fractional concentrations in the monounsaturated (PC34:1) and di-unsaturated species (PC34:2, PC36:2) (Figure 5). There were no time course variations in the fractional composition of the PC species, which remained stable during the 96 h study period for both cell types from both healthy controls and ARDS patients (Figure 5B,C). Overall, there were increments in all mono- and di-unsaturated PC species with significant decreases in saturated and polyunsaturated PC species. There were no differences in the 1-alkyl PC compositions between the groups (Figure 6).

### 3.5. PC Kinetics in CD15+ Cells in Healthy Humans and Patients with ARDS

The dynamic flux of individual PC molecular species in CD15+ cells through the CDP-choline pathway was quantified from the incorporation of *methyl*-D_9_-choline, determined by the precursor scan of *m/z* 193+ fragment ion derived from the headgroup of newly synthesised *methyl*-D_9_-choline-labelled PC (Figure 7). *Methyl-*D_9_ enrichments in the total PC and individual PC molecules were calculated as the abundance of labelled PC as a fraction of the total labelled and unlabelled PC at intervals up to 96 h after the *methyl*-D_9_ choline infusion.

The incorporation of *methyl-*D_9_ into CD15+ cells was rapid for both healthy volunteers and ARDS patients, with enrichments at the earliest time points of 0.54 ± 0.06% (t = 8 h) and 0.65 ± 0.08% (t = 6 h) of the total PC, respectively. This incorporation then increased more slowly for both groups, reaching maximal enrichments at t = 72 h of 0.75 ± 0.06% and 1.05 ± 0.13% total PC for healthy volunteers and ARDS patients, respectively (Figure 8A). The calculation of the rate of change clearly showed the bulk of label incorporation within the initial 24 h for both the patient and control groups (Figure 8B). Enrichment in the total PC was greater at each time point for ARDS patients. The incorporation pattern was not uniform between individual PC species, with more rapid rates in polyunsaturated PC species compared to those of all the other PC species for both the patient and control groups (Figure 8C,D respectively). An inspection of the enrichment patterns suggests that the incorporation into the polyunsaturated PC species might have been even more rapid than those shown in Figure 8C,D, but this could not be determined in the absence of earlier time points.

*Methyl-*D_9_ enrichment was greatest for PC36:4 at the earliest time points for both groups, followed by slow declines over subsequent days (Table 1). Label incorporation into di-unsaturated species was lower than for polyunsaturated PC species but followed a similar pattern with time. By contrast, incorporations into monounsaturated and alkyl PC species increased more slowly to reach maximal enrichments between 48 and 96 h. Despite these temporal variations, all PC molecular species achieved a state of equilibrium at 72 h after the *methyl-*D_9_-choline infusion (Figure 8C) for patients in the ARDS group but, in the healthy control group, *methyl-*D_9_ enrichment in polyunsaturated PC remained higher than other species even at 96 h (Figure 3). ARDS patients at the time of maximal enrichment had increased *methyl-*D_9_ enrichments compared with the control group for all PC molecular species except for the PUFA-PC species (Table 1).

### 3.6. Acyl Remodelling of Newly Synthesised Phosphatidylcholine in CD15+ Cells

The pattern of incorporation of *methyl*-D_9_ choline into PC molecular species at the earliest time point differed from that of the endogenous PC composition. For instance, newly synthesised PC contained more polyunsaturated molecular species than were present in the fractional composition of unlabelled PC (Figure 9). The contribution of polyunsaturated PC species to newly synthesised PC then subsequently decreased with time for both the control and ARDS groups. The fractional distribution of *methyl*-D_9_ choline incorporation into di-unsaturated PC species essentially mirrored that of polyunsaturated species for the healthy controls, but not for ARDS patients. This was partly due to the increased fractional concentration of di-unsaturated PC species in the ARDS patients. By contrast, monounsaturated and alkyl species were under-represented in newly synthesised PC species for both the control and ARDS groups but increased with time to reach equilibrium with the unlabelled PC compositions by 72–96 h.

## 4. Discussion

This study demonstrates, for the first time, not only that CD15+ and CD3+ cells maintain distinct molecular compositions of PC in vivo in healthy people (Figure 2), but also that alterations observed in ARDS patients are cell-type specific. As we previously reported (5), ether-linked and monounsaturated species were predominant in CD15+ cells, while significantly elevated concentrations of polyunsaturated species were observed in CD3+ cells. The CD15+ PC alterations in ARDS were limited mainly to decreased PC36:1 and increased PC36:2, with increases in the overall di-unsaturated and polyunsaturated PC fractional compositions. In contrast, CD3+ cells from ARDS patients showed increments in mono-unsaturated and di-unsaturated PC fractions with markedly decreased fractional concentrations of polyunsaturated PC species. The major factor here was a significant loss in arachidonoyl PC species (PC36:4 and PC38:4) in CD3+ cells from ARDS patients. The mechanistic reason behind these selective changes is not clear, but it appears that arachidoyl PC species are preserved in the CD15+ cells from ARDS patients compared with the CD3+ cells, although present at much lower fractional concentrations. This may be linked to the requirement for arachidoyl fatty acid for the synthesis of downstream lipid mediators through activated neutrophils. We also reported a loss in arachidonoyl PC in red blood cells from ARDS patients [11]. Our results in this study confirm that arachidonate loss is not only limited to the red blood cells, but also indicates that both peripheral red blood cells and CD3+ T-lymphocyte PC may act as arachidonate donors during acute inflammation.

The CDP-choline pathway is the primary source of PC synthesis in all mammalian cells except for the erythrocytes [13]. However, the molecular specificity of the final phospholipid membrane composition is largely dependent on the cell type and the capability for acyl-remodelling mechanisms by membrane-bound phospholipases. Phospholipid turnover in cellular membranes is rarely reported in humans, particularly humans in disease states. Moreover, most studies were performed on cell culture lines in vitro via gas chromatography to assess total esterified fatty acids and provided minimal information regarding PC molecular specificity and metabolism [14,15,16]. Similar to our findings in vivo, when whole human blood was incubated with *methyl*-D_9_ choline for 3 h, there was an increased turnover in arachidonoyl species and much slower enrichment in ether-linked species in neutrophil PC [5]. In the current study, we extended this observation by quantifying the synthetic patterns of individual PC molecules in vivo by infusing *methyl-*D_9_ choline and taking samples at serial time points. *Methyl*-D_9_ choline enrichment was 0.54% of the CD15+ cellular PC at the earliest time point in healthy controls, while corresponding enrichments were greater at all time points in cells from ARDS patients. Despite their higher enrichment, the peak *methyl-*D_9_ PC enrichments for polyunsaturated PC species were proportionately lower compared to all other PC species (Table 1), suggesting that polyunsaturated-based PC fractional synthesis is reduced in CD15+ during ARDS.

The composition of the newly synthesised PC fraction from CD15+ cells differs from that of the endogenous unlabelled fraction. There were much higher proportions of polyunsaturated and di-unsaturated species in the newly synthesised PC fraction at earlier time points, which subsequently equilibrated with the endogenous unlabelled PC fraction after 48–72 h. By 96 h, the labelled PC fraction was similar to that of the unlabelled PC endogenous composition. This suggests that there is specific molecular specificity toward structural membrane PC synthesis and acyl-remodelling mechanisms from the sequential actions of phospholipase and acyltransferase enzymes to maintain the final membrane PC composition. This finding is similar to those of previous studies demonstrating the phospholipase medicated acyl-remodelling mechanisms of dipalmitoyl-PC (PC32:0) synthesis and secretion from human and mice alveoli [17,18].

Although there were no significant changes to the CD15+ PC molecular species composition over time in ARDS patients, there was a continuous enrichment even 4 days after the *methyl-*D_9_ choline infusion. This suggests continued PC synthesis in CD15+ cells for a prolonged time, and it apparently conflicts with the established paradigm of a very short half-life for neutrophils. Although, historically, the reported plasma half-life of neutrophils in vivo is short, recent isotope-labelling studies using deuterated water and glucose suggest a variation in a circulating half-life between 19 h and 3.7 days [19,20]. One possible explanation for this continued enrichment is that *methyl*-D_9_ choline labelling may have started in the bone marrow and continued during the maturation and transit phases. The labelling can occur only in the cells that are alive, as there is continued enrichment suggesting PC synthesis via the CDP- choline pathway.

Neutrophils play a critical role in the development and progression of ARDS. There is a significant body of evidence suggesting their involvement in the pathogenesis of acute lung injury [8,21,22] and histological and radiological evidence of neutrophil accumulation in the lungs during ARDS [21,23]. The number of neutrophils and their secretory materials are increased in the bronchoalveolar lavage of patients with ARDS [24]. CD15+ cells are primarily neutrophils and eosinophils, with neutrophils being the predominant cell type. We found that there is a rapid flux of *methyl*-D_9_ choline enrichment in arachidonoyl PC species in CD15+ cells but with a proportionately lower composition, suggesting increased metabolism and turnover. Free arachidonate is liberated through the hydrolysis of ester bonds linking 20:4 to the glycerophospholipid backbone and is the first stage of the formation of hydroxylated and prostaglandin metabolites. Arachidonic acid containing ether phosphatidylcholines might also serve as a precursor for eicosanoids and platelet-activating factor [25,26,27].

Currently, the clinical relevance of these metabolic changes and how these phospholipid changes alter cellular function is not fully understood. However, the composition of phospholipids and their dynamic alterations in these cells may be related to the regulation and fluidity of these phospholipids for the synthesis and metabolism of proinflammatory and anti-inflammatory mediators such as eicosanoids and resolvins, and the analysis of downstream metabolites in future studies is required to characterise these further [28,29]. ARDS remains a therapeutic challenge for intensivists. Therapeutic strategies are limited with no established pharmacotherapy that modifies the disease process. Nutrition plays a crucial role in the moderation of inflammation in several conditions, and cellular phospholipid compositions can be modified by external nutritional supplementation [30,31,32,33]. Compared to serial assessments of lipid composition or content, novel isotope-labelling techniques provide mechanisms for studying the cellular–molecular phospholipid metabolism in vivo following such nutritional interventions.

There are several caveats to these study findings. Due to small cell numbers, we were unable to analyse the *methyl*-D_9_ choline enrichment in CD3+ cells. Also, as we concentrated on the quantification of *methyl*-D_9_ choline in PC, we were unable to quantify other important phospholipids such as phosphatidylethanolamine and phosphatidylinositol, which are major sources of arachidonic acid. We are unsure that the phospholipid changes demonstrated here are specific to ARDS or common to all acute inflammatory conditions. Further exploration is required comparing critically ill ARDS patients with non-ARDS patients. Moreover, we did not perform a phospholipid analysis on the alveolar cellular compartment, which may have provided detailed membrane phospholipid changes more specific to ARDS. Also, as we did not analyse the phospholipid compositional variations among the different subsets of lymphocytes, we cannot comment on how these changes alter cellular function. Moreover, due to the small number of patients, we could not correlate individual phospholipid PC fractional concentration or synthesis with clinical variables such as markers of inflammation or oxygenation. Nevertheless, this is the first study to evaluate the phosphatidylcholine molecular characteristics of immune cells in vivo in ARDS. Further studies are necessary to provide more details on these findings, including the assessment of clinical correlations, quantification of downstream metabolites of arachidonate pathways and functional cellular modifications as a result from these membrane phospholipid changes.

## 5. Conclusions

The perturbation of neutrophil-mediated inflammatory response leads to exaggerated injury in ARDS. The inflammatory cell phospholipid composition is tightly regulated. In vivo studies investigating the cellular dynamic membrane phospholipid composition and alterations during disease states are lacking. This is the first study to perform a dynamic molecular characterisation of cellular membrane phospholipids in patients with ARDS. The PC molecular composition is different between the cells studied and showed alterations in ARDS. There was an increased *methyl*-D_9_ PC enrichment in CD15+ cells in ARDS. There is also evidence for the acyl-remodelling of CD15+ phospholipids. The molecular specificity of PC synthesis was variable between PC species. The rapid turnover of arachidonate in neutrophils is likely due to the increased metabolic demand and the need for the substrate precursor for synthesizing downstream metabolic pro- and anti-inflammatory pathways.

## Figures and Tables

**Figure 1 cells-13-00332-f001:**
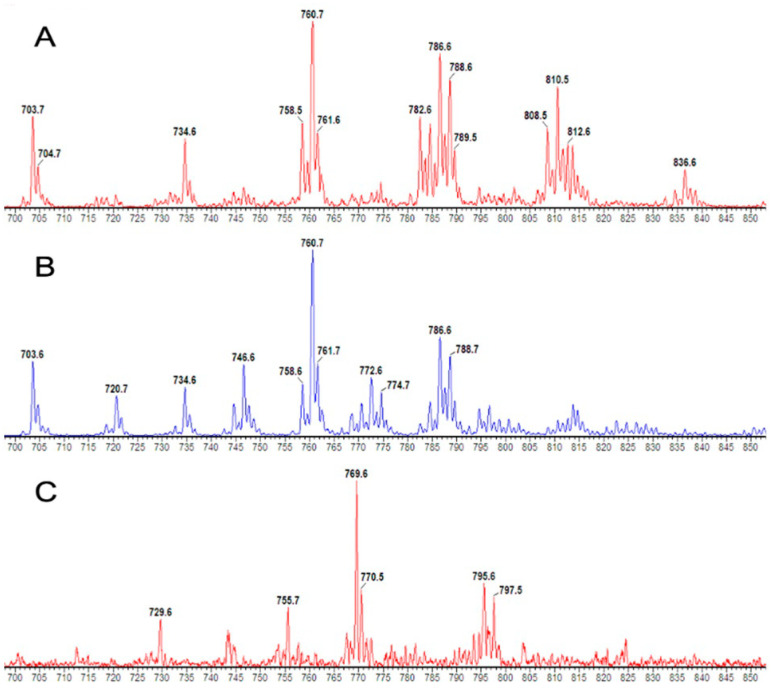
Parent scans of P184+ mass spectra for the PC composition of CD3+ (**A**), CD15+ (**B**) and P193+ for *methyl*-D_9_-labelled CD15+ PC composition (**C**).

**Figure 2 cells-13-00332-f002:**
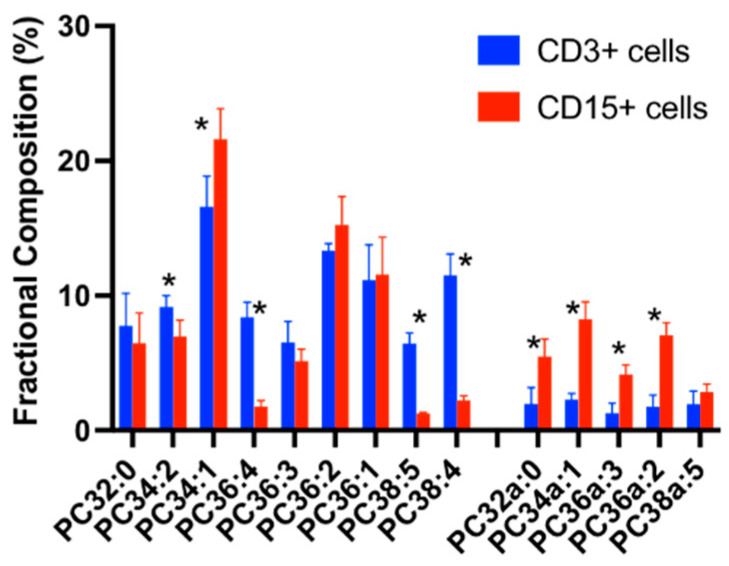
A comparison of the PC molecular compositions between CD3+ and CD15+ cells. Comparisons were conducted using the Student’s *t*-test and * *p* < 0.05.

**Figure 3 cells-13-00332-f003:**
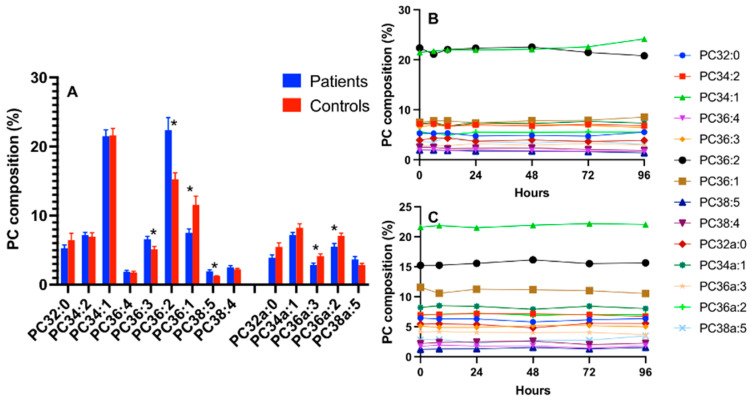
The PC compositions of CD15+ cells with a comparison between ARDS patients and healthy controls on enrolment (**A**) and the composition over time for the patients (**B**) and controls (**C**). Comparisons were conducted using the Student’s *t*-test and * *p* < 0.05.

**Figure 4 cells-13-00332-f004:**
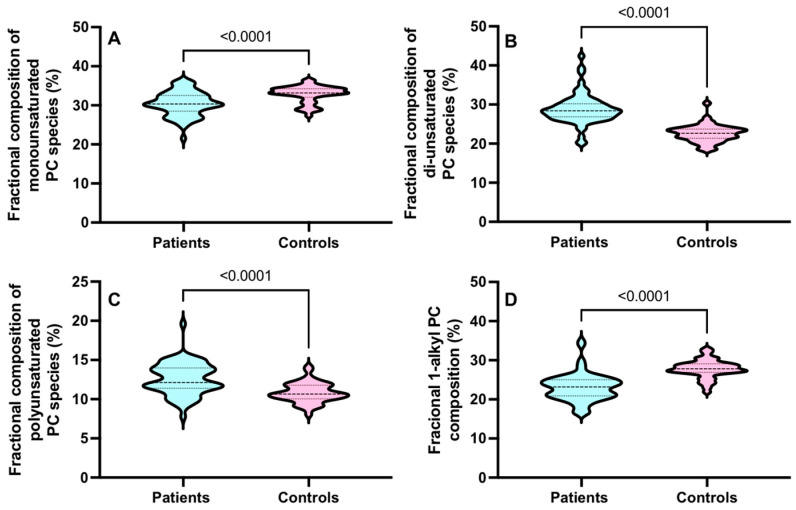
The CD15+ compositional variations between the monounsaturated (**A**), di-unsaturated, (**B**) polyunsaturated (**C**) and 1-alkyl PC (**D**) compositions in ARDS patients and controls for all sampling time points. Comparisons were conducted using the Student’s *t*-test.

**Figure 5 cells-13-00332-f005:**
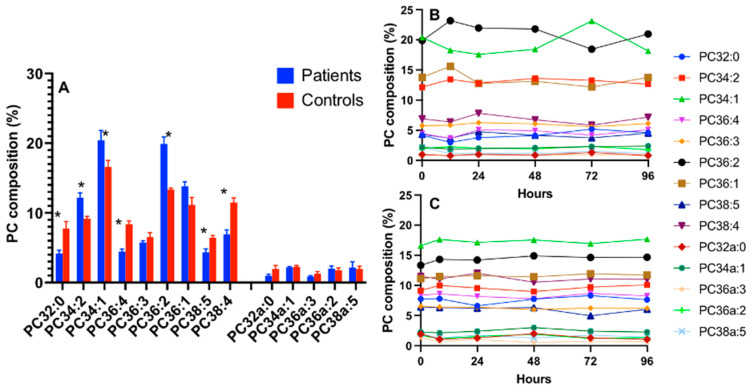
The PC compositions of CD3+ cells with a comparison between ARDS patients and healthy controls on enrolment (**A**) and the composition over time for the patients (**B**) and controls (**C**). Comparisons were conducted using the Student’s *t*-test and * *p* < 0.05.

**Figure 6 cells-13-00332-f006:**
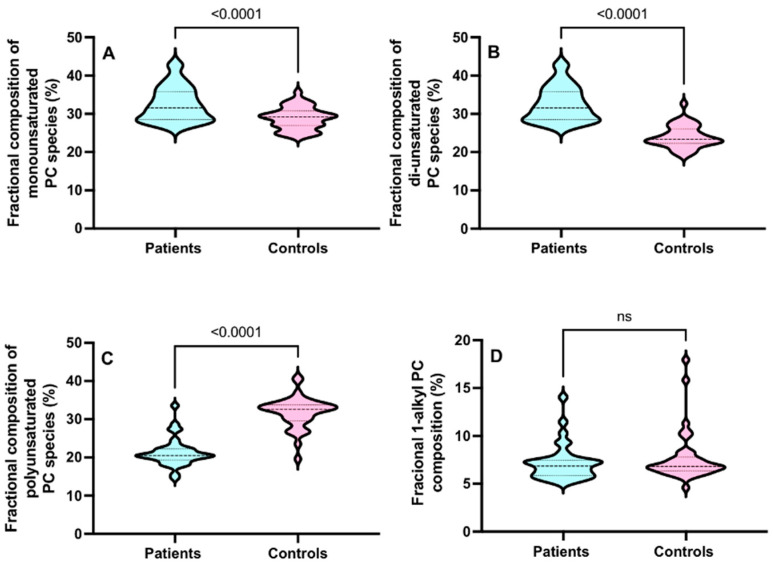
The CD3+ compositional variations between the monounsaturated (**A**), di-unsaturated, (**B**) polyunsaturated (**C**) and 1-alkyl PC (**D**) compositions in ARDS patients and controls for all sampling time points. Comparisons were conducted using the Student’s *t*-test.

**Figure 7 cells-13-00332-f007:**
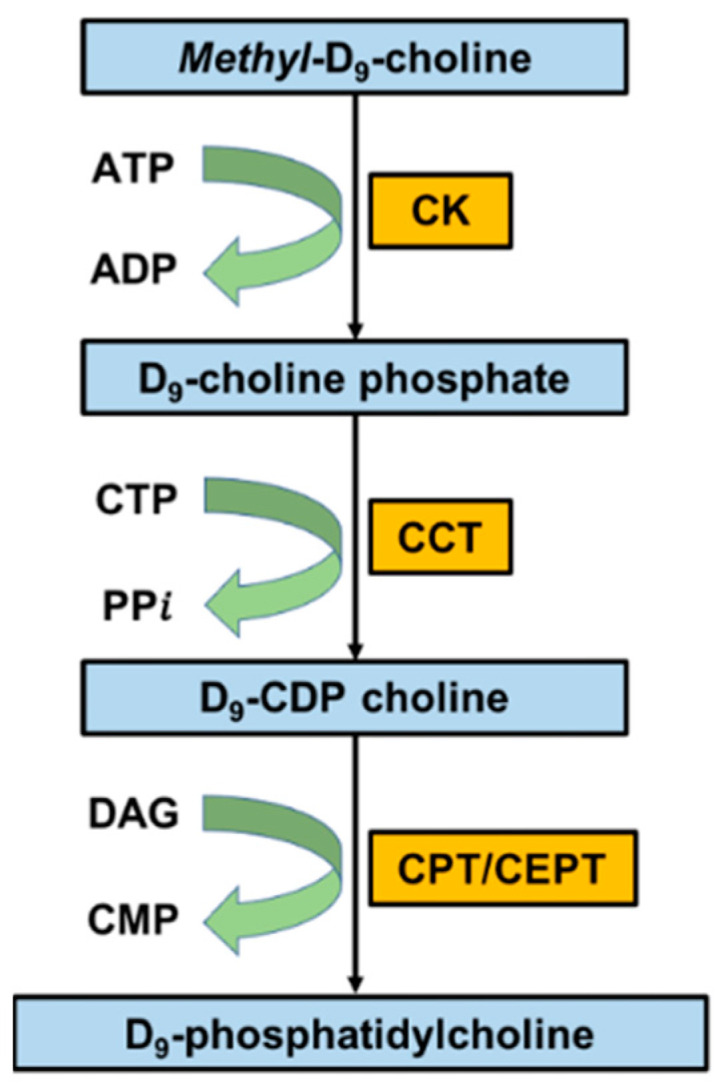
The enrichment of *methyl*-D_9_-choline into cellular PC through the CDP-choline pathway and the mass spectra for CD15+ cells with and without *methyl-*D_9_ enrichment.

**Figure 8 cells-13-00332-f008:**
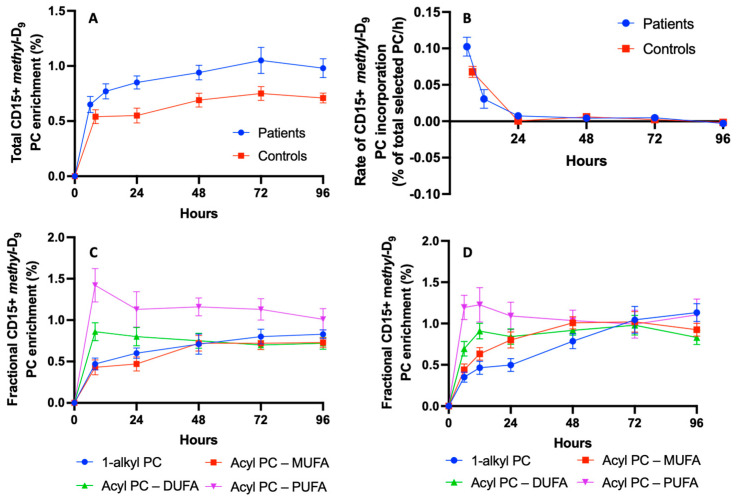
The *methyl*-D_9_ choline enrichment patterns in CD15+ cells. (**A**) *Methyl*-D_9_ choline enrichment comparison between patients and controls. (**B**) Rate of enrichment comparison between patients and controls. (**C**) Enrichment pattern for phospholipid species in healthy controls. (**D**) Enrichment pattern for phospholipid species in ARDS patients.

**Figure 9 cells-13-00332-f009:**
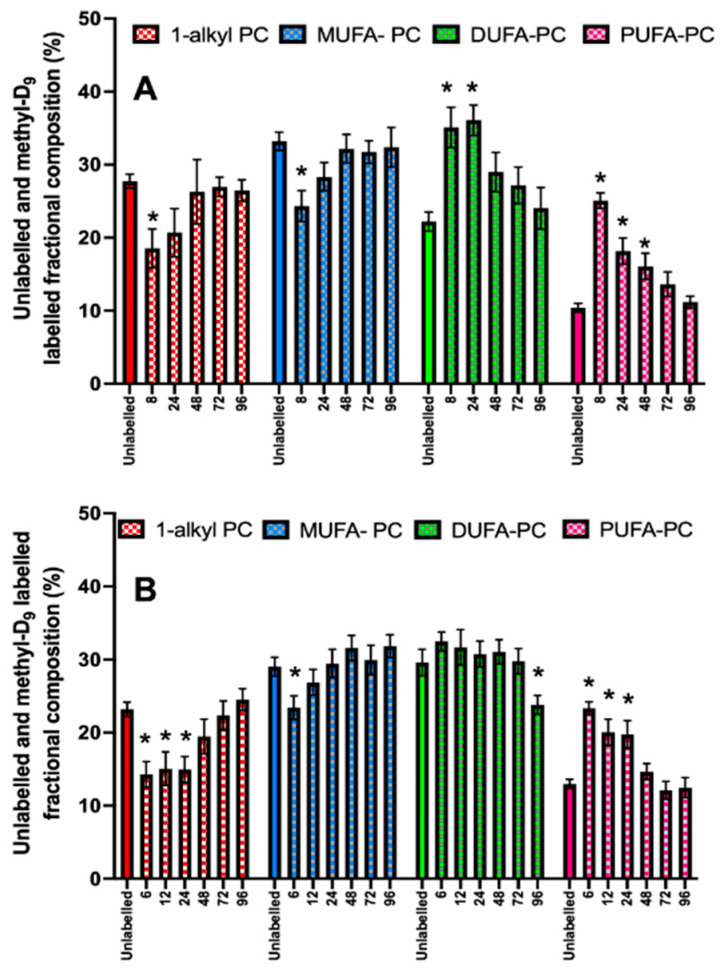
A comparison of unlabelled fractional compositions at Time = 0 with *methyl-*D_9_ labelled PC compositions over time for healthy controls (**A**) and ARDS patients (**B**). MUFA: monounsaturated fatty acid; DUFA: di-unsaturated fatty acid; PUFA: polyunsaturated fatty acid; PC: phosphatidylcholine. * *p* < 0.05 when compared with the unlabelled composition at T = 0.

**Table 1 cells-13-00332-t001:** The peaks and timings of molecular *methyl*-D_9_ PC enrichment and the percentage changes in enrichment in ARDS patients and healthy controls. * Data presented as mean (standard error of the mean).

PC Species	ARDS Patients		Controls		% Change in ARDS Patients
	Peak *Methyl*-D_9_ Enrichment (%) *	Time to Maximal Enrichment	Peak *Methyl*-D_9_ Enrichment (%) *	Time to Maximal Enrichment	
PC32a:0	1.34 (0.29)	72 h	0.97 (0.08)	96 h	+38.1%
PC32:0	1.38 (0.28)	72 h	1.07 (0.12)	72 h	+29.0%
PC34a:1	1.13 (0.12)	96 h	0.81 (0.07)	96 h	+39.5%
PC34:2	1.04 (0.14)	12 h	0.82 (0.12)	24 h	+26.8%
PC34:1	1.01 (0.12)	48 h	0.69 (0.03)	72 h	+46.4%
PC36a:3	1.17 (0.13)	96 h	0.82 (0.18)	72 h	+42.7%
PC36a:2	1.00 (0.10)	96 h	0.67 (0.05)	72 h	+49.3%
PC36:4	1.26 (0.25)	6 h	1.48 (0.18)	8 h	−14.9%
PC36:3	1.10 (0.11)	6 h	1.28 (0.23)	8 h	−14.1%
PC36:2	1.09 (0.12)	72 h	0.74 (0.08)	72 h	+47.3%
PC36:1	1.06 (0.16	72 h	0.81 (0.13)	48 h	+30.9%
PC38a:5	1.26 (0.16)	96 h	1.17 (0.35)	48 h	+7.7%
PC38:5	1.13 (0.19)	6 h	1.39 (0.34)	8 h	−18.7%
PC38:4	1.09 (0.12)	6 h	1.22 (0.29)	8 h	−10.7%

## Data Availability

The data that support the findings of this study are available upon request.

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
