# Peer review of "In Vivo Cellular Phosphatidylcholine Kinetics of CD15+ Leucocytes and CD3+ T-Lymphocytes in Adults with Acute Respiratory Distress Syndrome"

_cells, 2024, doi:10.3390/cells13040332_

Round 1

Reviewer 1 Report

Comments and Suggestions for Authors

This paper by Dushianthan  et al.  titled  In-vivo cellular phosphatidylcholine kinetics of CD15+ leucocytes and CD3+ T-lymphocytes in adults with acute respiratory distress syndrome” provides evidence of in vivo assessment  of  membrane composition and dynamic changes  in immunological cells from ARDS patients. The topic and the results are  interesting, particularly for their  implications in clinical  practice, given their support to preveious evidence  of alterations of  a simple and accurate prognostic  biomarker, such as neutrophil to lymphocyte  ratio (NLR),  of  Community Acquired Pneumonia (CAP) in the general population,  as well as of  survival and ICU admission in Covid-19 patients. Despite the very small number of patients, a distinction between direct (with epithelial  cell damage) and indirect  (with endothelial  cell damage) ARDS phenotypes  could  be interesting to identify  potential differences  in the cell phenotypes.

 Specific Comments

1.Prognostic role of NLR should  be brought to the fore, highlighting similarities with your findings and incorporating  references (PMID: 23049706, PMID: 35456328, PMID: 37373750), even emphasizing its role mirroring the derangement in the interplay between innate and adaptive immunity (PMID: 35408994). This would further strenghten the message of your findings to clinicians.

2. Is there any relationships between  the pattern of CD15+ and CD3+ T lymphocytes and CRP? These data would be interesting because it was recently shown by mediation analysis  in Covid-19 patients that neutrophils, CRP, and lymphocytes are significantly and directly linked to P/F, while the influence of inflammation on P/F, reflected by CRP, was also mediated by neutrophils (PMID: 37373750)

3.A recent report on metabolomics of ARDS (PMID: 3740804) identified a difference in ARDS subphenotypes. This issue is of prominent importance and  should be brought to the fore as compared to your findings, because identified  metabolites and pathways can provide clues relevant to the diagnosis and treatment of individuals with ARDS.

4. Information on ongoing  therapies would  help  to undesrtand whether the immunophenotype patterns  would  be strictly related to ARDS per se,  or could depend  on drugs administered in ICUs.

Author Response

We are grateful for the reviewer's comments, and please see our response to the comments.

Reviewer 1

This paper by Dushianthan  et al.  titled  “In-vivo cellular phosphatidylcholine kinetics of CD15+ leucocytes and CD3+ T-lymphocytes in adults with acute respiratory distress syndrome” provides evidence of in vivo assessment  of  membrane composition and dynamic changes  in immunological cells from ARDS patients. The topic and the results are  interesting, particularly for their  implications in clinical  practice, given their support to previous evidence  of alterations of  a simple and accurate prognostic  biomarker, such as neutrophil to lymphocyte  ratio (NLR),  of  Community Acquired Pneumonia (CAP) in the general population,  as well as of  survival and ICU admission in Covid-19 patients. Despite the very small number of patients, a distinction between direct (with epithelial  cell damage) and indirect  (with endothelial  cell damage) ARDS phenotypes  could  be interesting to identify  potential differences  in the cell phenotypes.

Specific Comments

1.Prognostic role of NLR should  be brought to the fore, highlighting similarities with your findings and incorporating  references (PMID: 23049706, PMID: 35456328, PMID: 37373750), even emphasizing its role mirroring the derangement in the interplay between innate and adaptive immunity (PMID: 35408994). This would further strengthen the message of your findings to clinicians.

Response: We have now introduced a sentence in the introduction section to incorporate the importance of the N/L ratio in clinical outcomes [lines 53-57]. Moreover, we did not have a milder group without ARDS to compare and explore the degree of NLR with these phospholipid changes.

  1. Is there any relationships between the pattern of CD15+ and CD3+ T lymphocytes and CRP? These data would be interesting because it was recently shown by mediation analysis in Covid-19 patients that neutrophils, CRP, and lymphocytes are significantly and directly linked to P/F, while the influence of inflammation on P/F, reflected by CRP, was also mediated by neutrophils (PMID: 37373750)

Response: This was a small study aimed at characterizing in vivo phospholipid composition and dynamic flux in ARDS. We would need a larger study to demonstrate variations in clinical outcomes with phospholipid composition and the dynamic flux of individual phosphatidylcholine species. We have introduced a sentence to address this in the limitations section [Line 613-615].  

3.A recent report on metabolomics of ARDS (PMID: 3740804) identified a difference in ARDS subphenotypes. This issue is of prominent importance and should be brought to the fore as compared to your findings, because identified  metabolites and pathways can provide clues relevant to the diagnosis and treatment of individuals with ARDS.

Response: These sub-phenotype profiles were based on cytokine measurements, including IL-6 and soluble tumour necrosis factor (sTNFr1), and we did not perform these in our study.

4. Information on ongoing  therapies would  help  to understand whether the immunophenotype patterns  would  be strictly related to ARDS per se,  or could depend  on drugs administered in ICUs.

Response: All patients had standard therapies with sedation and antibiotics, we did not administer any immunomodulating drugs, including corticosteroids. We agree with the reviewer that demonstration of changes specific to ARDS would require a comparison of ARDS patients with other critically ill patients who do not have ARDS. We agree that these changes may not be specific to ARDS, and we have introduced a sentence in the limitation paragraph to address this issue [line 606-608]. 

Reviewer 2 Report

Comments and Suggestions for Authors

- the authors could discuss what these changes in phospholipids influence on the cellular mechanism of ARDS

- the discussion needs more fluidity

Author Response

We are thankful for the reviewer's comments and please see our responses below.

Reviewer 2

- the authors could discuss what these changes in phospholipids influence on the cellular mechanism of ARDS

Response: We have introduced additional sentences to address this issue. Currently, there is poor understanding of how these changes alter cellular function. This would require additional cellular functional assessment. We have added a sentence in the discussion section [lines 272- 277 and 430-434].  

- the discussion needs more fluidity

Response: We acknowledge that the discussion section was somewhat disjointed, and it has been substantially rewritten and shortened.

Reviewer 3 Report

Comments and Suggestions for Authors

Dushianthan et al investigated blood samples of nine ventilated ARDS patients on intensive care and of five healthy controls for the composition and cell flux of phosphatidylcholines (PC) in PBMCs. The authors concentrated on CD3+ and CD15+ cells and demonstrated that ARDS patients showed significant changes to PC composition in their CD15+ cells. 

I have the following four major comments:

Re Disease Condition - ARDS is an acute respiratory condition and I would have expected the investigation of alveolar leucocytes instead or in addition to the PBMC experiments. The study cohort was mechanically ventilated and bronchoalveolar lavages were accessible. The question is why no CD3+/CD15+ cells of BAL lavages were investigated. Perhaps there is a caveat to investigating PC levels in other fluids than in blood I am not aware of. Given the trial efforts and expenditure, it will however not be possible to add the investigation of BAL CD3+/CD15+ cells so I suggest adding this limitation to the discussion.

Re Investigated Cells - A question is why T cells and myeloid cells were not further differentiated in FACS analyses before PC mass spectrometry to determine whether PC synthesis and turnover differed in the different cell subsets. The investigative material was per methods a 1ml blood sample and therefore would have likely not detected sufficient cell populations for further differentiation before mass spec. This seems the biggest issue here as further cell differentiation would perhaps have allowed a finer differentiation of PC changes. Would the authors please comment and add to the limitations if I have not overseen an explanation for this?

Re Patient Management – It would be important to see patient treatment details. The PC changes in neutrophils hint at an assumed longer half-life. External glucocorticoids are known to increase neutrophils in the blood by reducing migration out of the bloodstream and accelerating release from the bone marrow. I would propose adding a supplemental table with the drug management details of the nine ARDS patients. If available, I would check whether the administration of higher-dose steroids correlated with observed PC changes.

Re Observed PC changes - If indeed higher dose glucocorticoids were administered, it would be an important addition to demonstrate the GC influence on CD15+ PC composition ex vivo. I would suggest adding a subanalysis to further elucidate the cause of the PC changes in CD15+ cells.

Author Response

We are thankful for the reviewer's comments, and please see our responses below.

Reviewer 3

Dushianthan et al investigated blood samples of nine ventilated ARDS patients on intensive care and of five healthy controls for the composition and cell flux of phosphatidylcholines (PC) in PBMCs. The authors concentrated on CD3+ and CD15+ cells and demonstrated that ARDS patients showed significant changes to PC composition in their CD15+ cells. 

I have the following four major comments:

Re Disease Condition - ARDS is an acute respiratory condition and I would have expected the investigation of alveolar leucocytes instead or in addition to the PBMC experiments. The study cohort was mechanically ventilated and bronchoalveolar lavages were accessible. The question is why no CD3+/CD15+ cells of BAL lavages were investigated. Perhaps there is a caveat to investigating PC levels in other fluids than in blood I am not aware of. Given the trial efforts and expenditure, it will however not be possible to add the investigation of BAL CD3+/CD15+ cells so I suggest adding this limitation to the discussion.

Response: We agree with the reviewer; it would have been useful to compare cellular phospholipid composition and dynamic metabolism from the alveolar compartment. However, we did perform alveolar cell extraction during this investigation. We have added a sentence as a limitation [line 611-613].

Re Investigated Cells - A question is why T cells and myeloid cells were not further differentiated in FACS analyses before PC mass spectrometry to determine whether PC synthesis and turnover differed in the different cell subsets. The investigative material was per methods a 1ml blood sample and therefore would have likely not detected sufficient cell populations for further differentiation before mass spec. This seems the biggest issue here as further cell differentiation would perhaps have allowed a finer differentiation of PC changes. Would the authors please comment and add to the limitations if I have not overseen an explanation for this?

Response: We did not perform further cellular differentiation. It is possible to do this as mass spectrometry only requires small quantities of cellular material for compositional analysis, although more material is needed to analyse stable isotope label incorporation. This is something we are considering with the additional quantification of downstream pro and anti-inflammatory metabolites in future studies.

Re Patient Management – It would be important to see patient treatment details. The PC changes in neutrophils hint at an assumed longer half-life. External glucocorticoids are known to increase neutrophils in the blood by reducing migration out of the bloodstream and accelerating release from the bone marrow. I would propose adding a supplemental table with the drug management details of the nine ARDS patients. If available, I would check whether the administration of higher-dose steroids correlated with observed PC changes.

Response: No external glucocorticoids were given to these patients.

Re Observed PC changes - If indeed higher dose glucocorticoids were administered, it would be an important addition to demonstrate the GC influence on CD15+ PC composition ex vivo. I would suggest adding a subanalysis to further elucidate the cause of the PC changes in CD15+ cells

Response: We did not give any immunomodulating drugs to these patients. Currently, the assumption is that the composition and turnover of these phospholipids are related to the synthesis and metabolism of downstream metabolites more than any patient-specific factors. This requires detailed analysis of downstream metabolites, which was not done as part of the study.

Round 2

Reviewer 1 Report

Comments and Suggestions for Authors

No further concern.

Reviewer 3 Report

Comments and Suggestions for Authors

The authors have satisfactorily answered all my concerns. I have no further comments.

Comments on the Quality of English Language

No concerns